# Development, promotion, and application of online OvAge calculator based on the WeChat applet: Clinical prediction model research

**Wenwen Xu**[1], **Hui Wang**[2,3], **Linting Han**[3], **Xueli Zhao**[3], **Panpan Chen**[3], **Haiyang Zhao**[3,4], **Jing Jin**[1], **Zheng Zhu**[5], **Fang Shao**[6]*, **Qingling Ren**[1]*

**1** Department of Gynecology, Jiangsu Provincial Hospital of Traditional Chinese Medicine, Affiliated Hospital of Nanjing University of Chinese Medicine, Nanjing, China, **2** Experimental Teaching Center of Basic Medicine, Nanjing Medical University, Nanjing, China, **3** Department of Histology and Embryology, Nanjing Medical University, Nanjing, China, **4** Innovation Research Institute of Traditional Chinese Medicine, Shanghai University of Traditional Chinese Medicine, Shanghai, China, **5** School of Pediatrics, Nanjing Medical University, Nanjing, China, **6** Department of Biostatistics, School of Public Health, Nanjing Medical University, Nanjing, China

* yfy0047@njucm.edu.cn (QR); shaofang@njmu.edu.cn (FS)

**Data Availability Statement:** The raw data supporting the conclusions of this article will be made available by Medical Ethics Committee of Jiangsu Hospital of Traditional Chinese Medicine.

## Abstract

Ovarian age assessment is an important indicator to evaluate the ovarian reserve function and reproductive potential of women. At present, the application of ovarian age prediction model in China needs further improvement and optimization to make it more suitable for the actual situation of women in China. In this study, we collected subjects and their data in three ways: firstly, we collected clinical data from a number of women go to local hospital, including healthy women and women with DOR or PCOS; secondly, we obtained data by recruited healthy women through CRO companies for a fee; thirdly, we collected data from a number of healthy women using WeChat applet. Using the data collected by CRO company and WeChat applet, we applied the generalized linear model to optimize the ovarian age prediction model. The optimized formula is: OvAge = exp (3.5254–0.0001*PRL-0.0231*AMH), where $P = 0.8195$ for PRL and $P = 0.0003$ for AMH. Applying the formula to the hospital population data set for testing, it showed that the predicted ovarian age in the healthy women was comparable to their actual age, with a root mean squared error (RMSE) = 5.6324. The prediction accuracy was high. These data suggest that our modification of the ovarian age prediction model is feasible and that the formula is currently a more appropriate model for ovarian age assessment in healthy Chinese women. This study explored a new way to collect clinical data, namely, an online ovarian age calculator developed based on a WeChat applet, which can collect data from a large number of subjects in a short period of time and is more economical, efficient, and convenient. In addition, this study introduced real data to optimize the model, which could provide insights for model localization and improvement.

Since sensitive patient information is involved in the data, the data is only available upon request. The email address of Medical Ethics Committee of Jiangsu Hospital of Traditional Chinese Medicine is llwyhbgs@qq.com.

**Funding:** This work was supported by grants from Jiangsu Provincial Administration of Traditional Chinese Medicine (ZD201915, a major project of Jiangsu Province's traditional Chinese medicine science and technology development plan in 2019).

**Competing interests:** The authors have declared that no competing interests exist.

## Introduction

In reproductive medicine, a woman's fertility potential can be used to measure the likelihood of a successful pregnancy using assisted reproductive technology. Domestic and foreign scholars in the field of reproduction have proposed models of female ovarian scoring and staging systems for predicting pregnancy outcomes, such as What's My Fertility, Stages of Reproductive Aging Workshop (STRAW), OvAge, and the AFA (anti-Müllerian) Hormone level-follicle-stimulating hormone level-age) models [1–5]. These models can help women of childbearing age to effectively understand their fertility status so that they can reasonably plan for pregnancy. Due to confounding factors such as age, genetics, environment, psychology, society, lifestyle, iatrogenic infection, immunity, etc., the age of the ovaries of women of the same chronological age is also very different [6–11]. Moreover, the fertility and ovarian quality of women vary from country to country due to their differences in national conditions, regional characteristics, and living habits. Therefore, it is important to localize the existing female ovarian scoring model for predicting pregnancy outcome and transform it into a tool suitable for Chinese women [12].

Currently, indicators commonly used in clinical to assess ovarian age and pregnancy outcomes include age, anti-Müllerian hormone (AMH), atrial follicle count (AFC), follicle-stimulating hormone (FSH), clomiphene citrate stimulation test (CCST), inhibin B (INHB) or a combination of the above [13–17]. The ideal scoring system and model should be universally accepted, repeatable, easy to understand, and applicable to the general population. We selected the ovarian age calculator OvAge for localization improvement to make it more suitable for ovarian age prediction of Chinese women.

Initially, we used two traditional methods of recruiting subjects. In the first stage, we recruited women who visited the gynecology and reproductive medicine department of local hospital and obtained their clinical data. Data collected in this way were relatively reliable but time consuming. In the second stage, we recruited healthy women through CRO (Contract Research Organization) and monitored their relevant indicators. The obtained data are relatively reliable but time consuming, costly, and subjects did not cover all women. Therefore, we wanted to explore a method of collecting clinical data that could compensate for the shortcomings of the two methods.

With the rapid development of the internet and 5G, people's lives are inseparable from the internet. In the current open network environment, WeChat has become a phenomenon-level application software in China [18]. Taking advantage of WeChat's characteristics of fast information dissemination and strong interactivity, we developed an online OvAge calculator based on the WeChat applet by optimizing the existing OvAge model. To obtain the estimated value of ovarian age, the subject could directly search for the "ovarian age calculator" applet in WeChat and then enter the test interface. Subjects only need to enter the required values in the applet to obtain their predicted ovarian age or ovarian age and fertility assessment results and obtain professional medical advice.

This study used data collected in three ways to optimize the OvAge model for predicting ovarian age to make it more suitable for Chinese women and demonstrated the feasibility of this optimization through a series of data analyses. This study can provide some ideas for the localization improvement of the model and provide more data support for the further optimization of the OvAge age calculator.

## Materials and methods

### Choice of indicators

In data collection for this study, we collected the AMH, FSH, luteinizing hormone (LH), estradiol ($E_2$), progesterone (P), testosterone (T), prolactin (PRL), and vitamin D levels of subjects

from Day 2 to Day 4 of the menstrual cycle. At the same time, we collected pelvic or vaginalul-
trasound images on the same day of hormone testing and recorded the subjects' height and
weight. One special point is that pelvic or vaginal ultrasound was dispensable for subjects col-
lected from the WeChat applet.

## Subject's data collection

This process is divided into three stages: (1) Phase I data collection from May 1 to June 30,
2020. We collected clinical data from subjects from the Department of Gynecology and Repro-
ductive Medicine of Jiangsu Provincial Hospital of Traditional Chinese Medicine, including
health, PCOS, and DOR populations. Within two months, we recruited 113 women (16
healthy individuals, 45 PCOS, and 52 DOR populations). The study was approved by the
China Clinical Trials Registry (Registration Number ChiCTR2000037522) and the Medical
Ethics Committee of Jiangsu Hospital of Traditional Chinese Medicine (Application Number
2019NL-152-02). The experimental design fully considered the principles of safety and fair-
ness, and the study did not cause any harm or risk to the subjects. Recruitment was based
entirely on the principle of voluntary and informed consent, and the privacy of volunteers was
protected as much as possible. In addition, there were no conflicts of interest in this study, (2)
Phase II data collection occurred from July 1st to August 31st, 2020, and the subjects were
recruited through a contract research organization (CRO) company for a fee. A total of 52
healthy women were included. All tests were carried out in the laboratory of Jiangsu Provincial
Hospital of Traditional Chinese Medicine, (3) Phase III data collection occurred due to the
low collection efficiency in the first two stages. We tried a brand new online OvAge calculator
with a subject self-report system based on a WeChat applet. We clarified the functions and
requirements of the OvAge calculator and handed it over to a professional software develop-
ment company to complete using JavaScript and WXML (WeiXin markup language) develop-
ment languages, and the basic software version 1.0.0 launched on September 23rd, 2020. In
two weeks, more than 9,000 sample data points were collected using the online OvAge calcula-
tor in the WeChat applet. After screening, 97 cases of healthy population data were available.

## Subjects

The age range of patients participating in this study through the above three recruitment meth-
ods was 15–55 years old.

The training set object used to build the model was healthy women. The inclusion criteria
for healthy women were a history of spontaneous conception, intact ovaries, and normal men-
struation, with an average interval of 21 to 35 days. The exclusion criteria were estrogen or
progesterone use or breastfeeding two months before registration, pregnancy, history of female
infertility, endometriosis, ovarian cystic mass, history of ovarian surgery, PCOS, gynecological
malignancy, history of radiotherapy or chemotherapy, autoimmune diseases, chronic systemic
metabolic and endocrine diseases including hyperandrogenemia, hyperprolactinemia, diabe-
tes, thyroid disease, hypogonadotropic hypogonadism, and drug usage that may cause irregu-
lar menstruation.

In addition, the test set also included PCOS and DOR populations. PCOS was diagnosed
according to Rotterdam criteria, and there were at least 2 of the following 3 characteristics:
hypomenorrhea/amenorrhea, clinical and/or biochemical hyperandrogenemia, and polycystic
ovarian morphology (PCOM) [19]. The diagnosis of DOR has not been standardized, and
there is no international standard. Combined with the current guidelines of the National ART
Surveillance System (NASS) and Society for Assisted Reproductive Technology (SART),
the diagnostic criteria used in this study were age $\geq$ 40 years old, basic FSH $\geq$ 12 mIU/mL,

AFC $\leq$ 5~7, and AMH $\leq$ 1.1 ng/mL. Meeting two or more of the above four items was determined to be DOR [20].

In this study, subjects from gynecology and reproductive medicine clinics, CRO companies, and OvAge WeChat applets were screened based on the above criteria, and past medical history was collected through professional questionnaires.

## Statistical analysis, model construction, and optimization

R software (version 4.0.2) was used to analyze a dataset of 262 subject records in the first to third phases. The whole analysis consists of 4 main steps (Fig 1). Flow chart of optimizing the OvAge model.

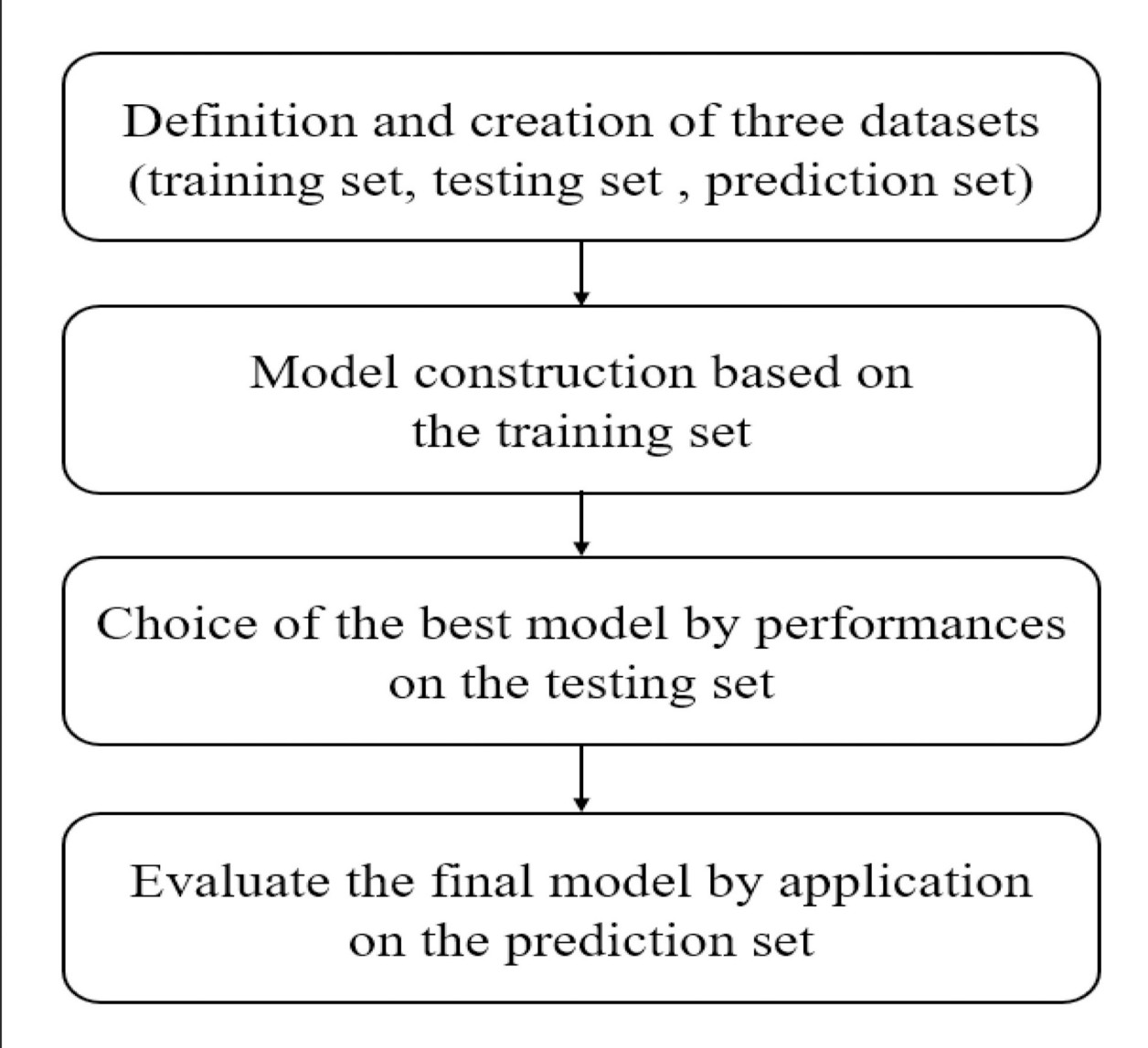

**Fig 1. Flowchart for optimizing the OvAge model.**

1. First, we divide the data collected in the three stages into three datasets: training set, test set, and prediction set. The training set includes phase II and phase III data, which include 52 and 97 observation data points, respectively. The prediction set included phase I data, including healthy controls (HC,16 subjects), DOR (52 subjects), and PCOS (45 subjects). We checked each dataset to ensure data quality, including missing values and consistency. For missing data, discard the corresponding observations and then perform modeling.

2. To describe the relationship between the new response variable, OvAge, and the set of inputs, which are the independent variables, we applied generalized linear model (GLM) theory to our research for the training set. The GLM provided a unified method for modeling all types of response variables, such as continuous, binary, and ordinal response or variables in the form of proportions. It has been hypothesized that in the healthy population, OvAge is equal to chronological age, which is not continuous and nonnormal. Therefore, a Poisson distribution was chosen as the random component of the GLM for modeling the expected value of OvAge. Identity and logarithm were chosen as candidate link functions. For Ockham's razor (law of parsimony) and avoiding collinearity problems due to the possible nonindependence of predictor variables, stepwise selection, a conventional variable selection method, was applied using the Akaike information criterion (AIC). We first constructed regression models for only Phase 2 data. Next, we used the same variables in the models for only Phase 2 data to fit corresponding models for both Phase 2 and 3 data.

3. Considering all possible interactions among covariates to assess any improvements in AIC value and accuracy, the best model was rebuilt. Interaction terms in the model provide interactive effects and were considered when the main effects were significant. The final model was chosen by comparing pairs of models for only Phase 2 data and both Phase 2 and 3 data with the assessment of prediction indices, mean squared error (MSE), root mean squared error (RMSE), and mean absolute error (MAE), for the test set of 19 observations. (iv) The final generalized linear model was applied to the healthy controls (HCs), DOR, and PCOS datasets in the prediction set. The hypothesis was that, in the case of HCs, the ovarian age would be equal to the chronological age. In the case of DOR, the ovarian age would be higher than the chronological age, and in the case of PCOS, the ovarian age would be lower than the chronological age.

## Results

### The OvAge calculator in the literature is not fully adapted to the Chinese population

According to the OvAge calculator in the literature, during phase I data collection in this study, we collected the chronological age of 113 women (16 healthy individuals, 45 PCOS, and 52 DOR patients) with five indicators related to the formula (Table 1).

We used the OvAge calculator in the literature to process the Phase I data and verified their biological age with the predicted ovarian age. The chronological age of PCOS patients was significantly older than the ovarian age calculated by OvAge ($p < 0.001$). The chronological age of DOR patients was significantly younger than the ovarian age calculated by OvAge ($p < 0.05$). There was no significant difference between the chronological age of healthy individuals and the ovarian age calculated by OvAge, but the P value was close to the critical value, suggesting a trend of difference (Table 2). The above results indicate that the chronological age for healthy people is not completely consistent with the ovarian age calculated by OvAge in the literature.

**Table 1. Participant data from Phase I.**

| Indicators | Healthy Women (n = 16) | DOR Women (n = 52) | PCOS Women (n = 45) |
|---|---|---|---|
| Chronological Age | 28.44±2.99 | 29.27±3.76 | 28.33±2.83 |
| AMH (ng/mL) | 3.64±3.42 | 1.02±1.03 | 12.79±3.43 |
| FSH (mIU/mL) | 6.28±2.00 | 22.70±11.64 | 5.80±1.87 |
| PRL (ng/mL) | 14.28±5.13 | 12.43±4.51 | 10.72±4.78 |
| AFC | 5.63±2.09 | 3.42±1.39 | 9.14±4.37 |
| BMI | 23.39±4.22 | 23.36±2.81 | 26.58±6.41 |

DOR: Diminished ovarian reserve. PCOS: Polycystic ovary syndrome. AMH: Anti-Müllerian hormone. FSH: Follicle-stimulating hormone. PRL: Prolactin. AFC: Antral follicle counting. BMI: Body mass index. n = 16: 16 sample data for healthy women. n = 52: 52 for DOR women. n = 45: 45 for PCOS women.

## The establishment of the generalized linear model for the new OvAge calculator

For healthy people, the chronological age was not exactly the same as the ovarian age calculated by OvAge in the literature, suggesting that this calculator is not completely suitable for Chinese women. To establish a more suitable ovarian age assessment method for the Chinese population, we tried to optimize the OvAge formula. Age was the dependent variable, and the independent variables for modeling were BMI, P, T, $E_2$, PRL, LH, FSH, and AMH. We considered normal linear regression, a combined generalized linear model with logarithm and identity link functions, and modeled it with the independent variable of age. We used stepwise linear regression based on the AIC value for variable selection and constructed a regression model for the phase II data, which are represented as mod1a, mod2a, and mod3a (Mod1 indicates that the dataset conforms to the normal distribution. Mod2 indicates that the dataset is a nonnormal Poisson distribution, and the GLM with the conventional log-link is selected. Mod3 indicates that the dataset is a nonnormal Poisson distribution and the GLM with the identity-link is selected. The "a" indicates that the selected dataset is from dataset II, and "b" indicates that the selected dataset is from dataset II and dataset III. "a" indicates that the selected dataset is from dataset II. Mod1a, mod2a, and mod3a only contain PRL and AMH variables. PRL was not statistically significant, whereas AMH was statistically significant (Table 3).

According to the results of prediction errors Mean Squared Error (MSE), Root Mean Squared Error (RMSE), and Mean Absolute Error (MAE) (Table 4), the smaller the MSE, RMSE, and MAE, the better the prediction. It was concluded that mod3a was the best fit to phase II data.

We used the OvAge calculator from the literature and the established mod3a formula to test the phase I and phase II datasets to compare the chronological age with the predicted ovarian age (Table 5). The Mod3a calculation results show that there was no significant difference

**Table 2. Comparison of chronological age and age calculated by OvAge in the literature.**

| Indicators | Chronological Age | Literature OvAge | *P* value |
|---|---|---|---|
| Healthy women (n = 16) | 28.44±2.99 | 30.63±3.02 | 0.0818 |
| DOR women (n = 52) | 29.27±3.76 | 31.29±5.42 | 0.0402 |
| PCOS women (n = 45) | 28.33±2.83 | 26.55±2.46 | 0.0002 |

n = 16: 16 sample data for healthy women. n = 52: 52 for DOR women. n = 45: 45 for PCOS women.

**Table 3. Summary of model fit for Phase II.**

| Item | | Estimate | Std. Error | T value | *P* value |
|---|---|---|---|---|---|
| Mod1a | (Intercept) | 32.3968 | 1.9030 | 17.0244 | <0.0001 |
| | PRL | 0.1483 | 0.0872 | 1.7001 | 0.0955(-) |
| | AMH | -0.8752 | 0.3296 | -2.6557 | 0.0106(*) |
| Mod2a | (Intercept) | 3.4846 | 0.0568 | 61.4003 | <0.0001 |
| | PRL | 0.0043 | 0.0024 | 1.7452 | 0.0810(-) |
| | AMH | -0.0287 | 0.0103 | -2.7813 | 0.0054(**) |
| Mod3a | (Intercept) | 32.2573 | 1.7938 | 17.9829 | <0.0001 |
| | PRL | 0.1438 | 0.0867 | 1.6577 | 0.0974(-) |
| | AMH | -0.8163 | 0.2971 | -2.7474 | 0.0060(**) |

*** P < 0.001;

** P < 0.01;

* P < 0.05;

¯ P>0.05.

**Table 4. Comparison of fit results for phase II.**

| Models | MSE | RMSE | MAE |
|---|---|---|---|
| Mod1a | 69.2735 | 8.3231 | 5.3830 |
| Mod2a | 117.1521 | 10.8237 | 5.9000 |
| Mod3a | 67.7729 | 8.2324 | 5.3772 |

MSE: mean squared error. RMSE: root mean squared error. MAE: mean absolute error.

between the chronological age of healthy women and the ovarian age predicted by OvAge, and P2 was less than P1, indicating that the difference between the predicted results for healthy people was more significant than that of the OvAge calculator in the literature. The chronological age of PCOS patients was not significantly different from the ovarian age calculated by OvAge (p>0.05). The chronological age of DOR patients and the ovarian age calculated by OvAge were also not significantly different (p>0.05). This finding shows that the accuracy of Mod3a's prediction of ovarian age was worse than that of the literature OvAge calculator.

## Optimization of the generalized linear model of the OvAge calculator

To further optimize the OvAge calculation formula we established by expanding the sample size of the test data, we tentatively developed an online OvAge calculator based on the WeChat

**Table 5. Comparison of chronological age, OvAge calculated age in the literature, and the established Mod3a calculated age.**

| Indicators | Chronological Age | Literature OvAge | Mod3a | *P1* value (literature) | *P2* value(Mod3a) |
|---|---|---|---|---|---|
| Healthy women (n = 68) | 30.79±5.91 | 31.45±4.89 | 31.44±2.27 | 0.3985 | 0.3344 |
| DOR women (n = 52) | 29.27±3.76 | 31.29±5.42 | 28.17±2.42 | 0.0402 | 0.1097 |
| PCOS women (n = 45) | 28.33±2.83 | 26.55±2.46 | 28.66±1.60 | 0.0002 | 0.4984 |

n = 68: 68 sample data for healthy women. n = 52: 52 for DOR women. n = 45: 45 for PCOS women.

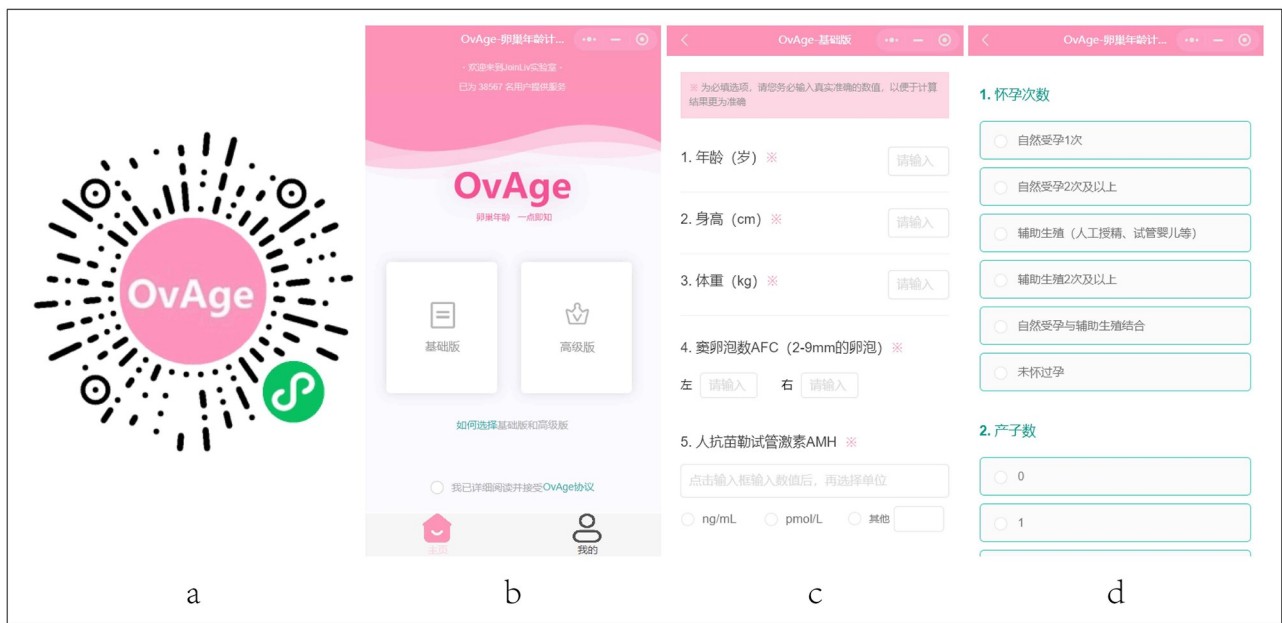

**Fig 2. Self-reporting system for subjects based on WeChat applet-the online OvAge calculator.** (A) QR code of online OvAge calculator. (B) Main interface. (C) Data Acquisition Module. (D) Questionnaire module.

applet using the mod3a formula (Fig 2). A subject self-report system based on the WeChat applet-online OvAge calculator. a. QR code of online OvAge calculator; b. Main interface; c. Data Acquisition Module; d. Questionnaire module; With the help of the online OvAge calculator in the WeChat applet, 9000+ sample data were collected in only two weeks, and data from 97 healthy people were screened out as model optimization dataset III.

As mentioned earlier, we used the data from phase II and phase III, and based on mod1a, mod2a, and mod3a, the models were optimized to mod1b, mod2b, and mod3b (b means the selected data came from the sum of phase II data and phase III data). We corrected the differences between datasets from different sources by introducing a new variable New (New = 0 represents phase II data, New = 1 represents phase III data) and considered adding new variables and their corresponding interaction terms with other variables to this model. However, the results of variable selection based on the stepwise linear regression of the AIC value showed that no significant variable effects related to the new variables were found, which means that there was no difference in AMH and PRL between the phase II and phase III data(Table 6). At the same time, the prediction errors of MSE, RMSE, and MAE were the lowest in mod2b (Table 7), indicating that mod2b was the best formula after the second optimization. The model was estimated as:

$$\text{OvAge} = \exp\left(3.5254 - 0.0001 * \text{PRL} - 0.0231 * \text{AMH}\right).$$

We used the literature formula, the best formula from the first optimization (mod3a), and the best formula from the second optimization (mod2b) to test with the phase I+II dataset (Table 8). The test results show that for the disease population (DOR/PCOS), P1<P3<P2, and the accuracy of the calculation results of the ovarian formula after two optimizations was not as good as that of the literature formula, but the accuracy of the calculation results of the second optimization formula was significantly improved. At the same time, for healthy people,

**Table 6. Summary of model fit for Phase II and Phase III.**

| Item | | Estimate | Std. Error | T value | P value |
|------|------|---------|-----------|---------|---------|
| **Mod1b** | **(Intercept)** | 33.8402 | 0.8489 | 39.8635 | <0.0001 |
| | **PRL** | -0.0016 | 0.0072 | -0.2253 | 0.8221(-) |
| | **AMH** | -0.7045 | 0.1968 | -3.5794 | 0.0005(***) |
| **Mod2b** | **(Intercept)** | 3.5254 | 0.0264 | 133.3673 | <0.0001 |
| | **PRL** | -0.0001 | 0.0002 | -0.2282 | 0.8195(-) |
| | **AMH** | -0.0231 | 0.0064 | -3.6403 | 0.0003(***) |
| **Mod3b** | **(Intercept)** | 33.6966 | 0.8367 | 40.2746 | <0.0001 |
| | **PRL** | -0.0017 | 0.0071 | -0.2407 | 0.8098(-) |
| | **AMH** | -0.6608 | 0.1872 | -3.5297 | 0.0004(***) |

*** $P < 0.001$;

** $P < 0.01$;

* $P < 0.05$;

⁻ $P > 0.05$.

**Table 7. Evaluation of model fit for Phase II and Phase III.**

| Models | MSE | RMSE | MAE |
|--------|-----|------|-----|
| **Mod1b** | 32.0149 | 5.6582 | 4.1787 |
| **Mod2b** | 31.7237 | 5.6324 | 4.1703 |
| **Mod3b** | 32.6351 | 5.7127 | 4.2063 |

MSE: mean squared error. RMSE: root mean squared error. MAE: mean absolute error.

P3>P1>P2, the calculation results of the first optimization formula were not as accurate as the literature formula, but after the second optimization, the accuracy of the calculation result was better than the literature formula. Mod2b had higher accuracy in predicting the ovarian age of the normal population, which was better than Mod3a and literature formulas. At the same time, the accuracy of predicting the ovarian age of the disease population was better than mod3a, but it was still not as good as the literature formula.

## Discussion

The age of human organs is not exactly the same as the person's chronological age, as is the case for ovaries [21, 22]. The reproductive aging process for women is related to factors such as age, genes, environment, stress, diet, and pathology. Therefore, the reproductive system of women of the same age can be different. In view of the differences, how to intuitively and

**Table 8. Comparison of chronological age and three different models' calculated age.**

| Indicators | chronological Age | Literature OvAge | Mod3a | Mod2b | P1 value (literature) | P2 value (Mod3a) | P3 value by (Mod2b) |
|------------|-------------------|------------------|-------|-------|------------------------|------------------|---------------------|
| **Healthy women (n = 68)** | 30.79±5.91 | 31.45±4.89 | 31.44±2.27 | 30.95±1.63 | 0.3985 | 0.3344 | 0.8169 |
| **DOR women (n = 52)** | 29.27±3.76 | 31.29±5.42 | 28.17±2.42 | 28.28±1.74 | 0.0402 | 0.1097 | 0.1069 |
| **PCOS women (n = 45)** | 28.33±2.83 | 26.55±2.46 | 28.66±1.60 | 27.63±1.10 | 0.0002 | 0.4984 | 0.4089 |

n = 68: 68 sample data for healthy women. n = 52: 52 for DOR women. n = 45: 45 for PCOS women.

accurately assess women's ovarian reserve, fertility, and menopause is worth considering. At present, there is no satisfactory model that can evaluate the status of the female reproductive system through a single index or a combined index. The ideal model should be able to integrate all relevant information to evaluate ovarian reserve qualitatively and quantitatively. In 2015, the "Ovage" proposed by Ventureira and others met the current requirements. However, due to its limitations in terms of geography, population, and sample size, this formula may not be fully applicable to Chinese women. Therefore, in this study, firstly, we developed an online OvAge calculator based on the WeChat applet by combining the OvAge calculation formula of Venturella R et al. The backend data showed that it could achieve the fastest collection of data for a population sample of 9000 cases in two weeks. Although these data were relatively raw and uncontrollable, it was still able to filter out 97 complete and compliant data, and the collection process is almost costless. There is no doubt that this method is more efficient, cheaper and safer than traditional methods of clinical data acquisition. Secondly, we collected a total of 262 cases of Chinese population data from hospital, CRO and WeChat applet, and divided them into training set and test set, and made an adaptation optimization of Chinese population data based on the OvAge formula of Venturella R et al. We established a normal linear regression model with the independent variable-age, and used stepwise linear regression based on AIC values for variable selection. The optimized formula OvAge = exp (3.5254–0.0001*PRL-0.0231*AMH) is currently an appropriate model for ovarian age assessment in healthy Chinese women, and the accuracy of calculation is better than the original literature formula, which suggests that our optimization for the literature formula is feasible and effective. However, there is still a limitation that the accuracy of ovarian age prediction for DOR/PCOS population is low, which is mainly due to the limitation of sample size as only healthy population data are included in the training set, while the test set contains both healthy and disease population data. We will continue to optimize the ovarian age calculation formula using the same way to improve the accuracy of prediction for the Chinese disease population.

Building an accurate scientific model that can quickly predict the age of the ovaries requires a large amount of sample data from subjects. The traditional methods of recruiting subjects mainly include research center recruitment, hospital department recruitment, and third-party CRO company cooperative recruitment. Usually, research centers or hospital departments must recruit subjects themselves, and therefore, it takes a long time to collect a large amount of sample data. Cooperation with third-party CRO companies is also very expensive in addition to being time-consuming. Moreover, CRO companies with a small scale, lack of influence, low visibility, and weak overall strength cannot achieve nationwide recruitment [23].

To overcome the shortcomings of traditional recruitment methods, we creatively developed an online OvAge calculator based on WeChat applets, which are widely used in China, to provide a reference value for ovarian age for the Chinese population. This research is also the first application of IOT technology to explore the possibility of a new type of self-report system for Chinese families. We combined the WeChat applet platform and the OvAge calculator by Ventureira. et al. to optimize a new online calculator. This online calculator can collect a large number of samples in a short time, which saves considerable time and greatly reduces the capital cost. At the same time, background operators can use the collected data to continuously optimize and iterate formulas. The subject's self-report system only needs to enter part of the values from the physical examination report, fill out a simple questionnaire, and sign the informed consent form, and then they can obtain their own estimated ovarian age and targeted fertility guidance. It not only ensures a wide range of subjects but also significantly improves the efficiency of the recruitment process and saves the cost of clinical recruitment. For the phase I data in this study, through hospital recruitment, we recruited 113 subjects (16 healthy individuals, 45 PCOS, and 52 DOR populations) within two months. For Phase II data, we

collected data on 52 healthy women in two months through cooperation with a CRO recruitment company at a cost of 52,000 yuan. For Phase III data, we used the online OvAge calculator of the WeChat applet to collect a total of more than 9,000 people within two weeks and obtained data on 97 healthy women after screening. In addition, the system is managed by authorized staff, and the network used is located in the hospital network environment and is specially encrypted.

Increasing evidence has shown that the construction of a predictive model of ovarian function based on healthy women's ovarian reserve is of great significance for the accurate assessment of ovarian reserve. There are significant ethnic differences in ovarian reserve and reproductive aging. Blacks, Hispanics, and Asians have significantly lower pregnancy and birth rates than Whites. This retrospective study based on a Chinese population has important implications for the development of ovarian age prediction models based on Chinese female characteristics. Compared with previous research results, this research not only optimized the OvAge model but also added family history, nutritional status, psychological factors, body mass index, and other indicators that are highly related to fertility (Fig 2). We used the WeChat applet OvAge calculator to collect a large amount of sample data to continuously optimize the OvAge model, and at the same time, we continued to update the subject's self-report system. Through model optimization, for the Chinese population, we deleted some analysis indicators to make the evaluation results for ovarian age more reliable. However, the online OvAge calculator also has some limitations. For example, subjects must obtain a physical examination report before entering a value, which causes some subjects to withdraw from the program when they find that there is no corresponding report. In addition, since the data are input by the subject autonomously, the authenticity of the data needs to be further determined. In the future, we will add image recognition functions and require subjects to upload hospital test reports, and then the computer will extract the values to increase the authenticity of the data. At the same time, our second optimized ovarian age calculation formula, mod2b, is more accurate than the first optimization formula, mod3a, for the three populations, and the accuracy of the ovarian age evaluation results for healthy people is also better than the literature OvAge formula. However, the accuracy of the prediction results for ovarian age for the PCOS and DOR populations is significantly worse than that of the literature OvAge formula. The possible reason for this phenomenon is that the training data (phase II & phase III) used in this study from the first optimization to the second optimization are only data from healthy people and do not include data from PCOS and DOR populations. In the future, with the continuous promotion of small programs, a large amount of healthy and nonhealthy population data will continue to be incorporated into the back-end system. As of the deadline for manuscript writing (August 26, 2021), the back-end system collected 28,616 samples from the population (Fig 1). We will continue to use newly collected population data to optimize the existing ovarian age calculation formula. In the near future, with the continuous optimization of the formula, we are confident in establishing a more scientific ovarian age assessment system based on the Chinese population.

In summary, the ovarian age calculator established using data from Chinese population samples can help Chinese women detect changes in ovarian reserve. In addition, it can also intervene in time to improve reproductive potential, predict the time of menopause, improve reproductive prognosis, and increase the success rate of assisted reproductive technology.

## Conclusion

This research combines the existing OvAge model to develop an OvAge online calculator based on the WeChat applet. Then, we collected basic data and gynecological examination

reports from Chinese women through an online application, aiming to explore new methods for subject self-report and provide more data support for further optimization of the OvAge model. This approach provides more support for the development and promotion of the online OvAge calculator, creates a new method of network-based recruitment of clinical subjects, and provides new ideas for clinical research.

## Acknowledgments

We would like to express deep appreciation for the support of Jiangsu Provincial Administration of Traditional Chinese Medicine.

## Author Contributions

**Data curation:** Wenwen Xu, Linting Han, Xueli Zhao, Haiyang Zhao.

**Formal analysis:** Wenwen Xu, Fang Shao.

**Methodology:** Wenwen Xu, Zheng Zhu.

**Software:** Wenwen Xu.

**Writing – original draft:** Wenwen Xu.

**Writing – review & editing:** Wenwen Xu, Hui Wang, Linting Han, Xueli Zhao, Panpan Chen, Haiyang Zhao, Jing Jin, Fang Shao, Qingling Ren.

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
