## [Decision Letter · Decision Letter 0]

14 Jul 2022

PONE-D-22-09299Development, promotion, and application of online OvAge calculator based on the WeChat applet: Clinical prediction model researchPLOS ONE

Dear Dr. Ren,

Thank you for submitting your manuscript to PLOS ONE. After careful consideration, we feel that it has merit but does not fully meet PLOS ONE’s publication criteria as it currently stands. Therefore, we invite you to submit a revised version of the manuscript that addresses the points raised during the review process.

We look forward to receiving your revised manuscript.

Kind regards,

Jing Zhang

Academic Editor

PLOS ONE

Journal Requirements:

This work was supported by grants from Jiangsu Provincial Administration of Traditional Chinese Medicine (ZD201915, a major project of Jiangsu Province's traditional Chinese medicine science and technology development plan in 2019).

We would like to express deep appreciation for the support of Jiangsu Provincial Administration of Traditional Chinese Medicine.

However, funding information should not appear in the Acknowledgments section or other areas of your manuscript. We will only publish funding information present in the Funding Statement section of the online submission form. 

This work was supported by grants from Jiangsu Provincial Administration of Traditional Chinese Medicine (ZD201915, a major project of Jiangsu Province's traditional Chinese medicine science and technology development plan in 2019).

7. Please include a separate caption for each figure in your manuscript.

**Comments to the Author**

1. Is the manuscript technically sound, and do the data support the conclusions?

Reviewer #1: Yes

Reviewer #2: Yes

2. Has the statistical analysis been performed appropriately and rigorously? 

Reviewer #1: I Don't Know

Reviewer #2: Yes

3. Have the authors made all data underlying the findings in their manuscript fully available?

Reviewer #1: No

Reviewer #2: Yes

4. Is the manuscript presented in an intelligible fashion and written in standard English?

Reviewer #1: Yes

Reviewer #2: Yes

5. Review Comments to the Author

Reviewer #1: Recommendation: Overall, the study is well conducted. The statistical analysis is appropriate. In my opinion, the above-mentioned manuscript needs major revision.

My specific comments are as follows:

1. The method section of the abstract needs more clarification regarding the eligibility criteria and etc…

2. Presentation of the result section of the abstract should be based on the statistical language.

3. In line 60, authors without any support (or references) said that many documents show that the reproductive potential of women gradually decreases with age.

4. What is the main gap in knowledge? The introduction section is toll long and it is not clearly presented.

5. In order to recognize the validity and reliability of data collection or measurements, please describe this section much more than before.

6. Which approaches were used for sampling?

7. The sample size is based on which formula?

8. Please describe any efforts to address potential sources of bias.

9. In this first paragraph of the discussion it is necessary to depict the main results of the study.

10. In discussion section lines (264-272, 306,) there is no citation or references.

11. Discuss limitations of the study, taking into account sources of potential bias or imprecision.

12. Discuss both the direction and magnitude of any potential bias.

13. The manuscript should be edited by a native English speaker.

14. Please add the footnote of tables. The tables are chaotic.

15. Please discuss and compare the findings of this study with relevant literature much more than the current version.

Reviewer #2: There are two objectives of this study. The first one is to incorporate the OvAge calculator into the popular WeChat applet platform. The authors did very well on this aspect because over 9000 sample data were collected in only two weeks and 97 229 healthy people were screened out as model optimization dataset III. This is its strong point.

The second goal is to optimize a new online calculator to meets the unique physiological characteristics of Chinese women. Results from this study showed that OvAge = exp (3.5254-0.001*PRL - 0.0231*AMH), is currently a suitable model for ovarian age assessment of healthy Chinese women. In general, this is an interesting study that applies current informative technology into ovarian reserve prediction. Enclosed are my comments.

1. The abstract was not representative of the study. Large number of contents regarding disease population (DOR/PCOS) were include in methods and results but was not mentioned at abstract. Please ensure that abstract and study contents are consistent.

2. The optimized ovarian age calculation formula, mod2b, is more accurate than the literature OvAge formula in evaluating ovarian age of healthy patient population. However, the real value of a scientific model is to accurately predict disease population. This is the weak point of this study that weakens the value of this study.

3. A reasonable experimental design is needed. The accuracy of ovarian age prediction for disease patient population (PCOS and DOR) was significantly worse than that of the literature OvAge formula. The underlying reason is that training data (phase II & III) used in this study from the first and second optimization were only from healthy people. If this is the case, prediction outcomes from such a model are unreliable. The whole manuscript needs to be re-written.

4. The collected data were divided into three datasets: training set, test set, and prediction set.

Different populations (healthy or DOR, PCOS) differed in each dataset. I would suggest the authors to focus on model establishment of healthy Chinese women instead of trying to cover both populations.

5. Page 24 Line 265-267: Please provide references of the previous studies.

6. PLOS authors have the option to publish the peer review history of their article (what does this mean?). If published, this will include your full peer review and any attached files.

Reviewer #1: No

Reviewer #2: **Yes: **Yimin Shu

---

## [Author Response · Author response to Decision Letter 0]

8 Oct 2022

Reviewer 1:

1. The method section of the abstract needs more clarification regarding the eligibility criteria and etc…

Re: Thank you for your question, and we apologize for not being able to provide a detailed explanation of the eligibility criteria and etc., in the methods section of the original abstract. The specific eligibility criteria for data screening are described in detail in the subject section of the original materials methods. We have also revised the abstract accordingly to provide a more detailed explanation of the study methods. We hope that the revised abstract will present our results better.

2. Presentation of the result section of the abstract should be based on the statistical language.

Re: Thank you for your suggestion. We are sorry that our first draft was not clear enough in the abstract section to describe the results. We have made a comprehensive revision of the abstract and hope that the revised abstract will present the focus of the article better.

3. In line 60, authors without any support (or references) said that many documents show that the reproductive potential of women gradually decreases with age.

Re: Thank you for pointing out the problem. We have added references to support this argument: Balasch J, Gratacós E. Delayed childbearing: effects on fertility and the outcome of pregnancy. CurrOpinObstet Gynecol. 2012;24(3):187-193. doi: 10.1097/GCO.0b013e3283517908

4. What is the main gap in knowledge? The introduction section is toll long and it is not clearly presented.

Re: Thank you for your comment. We believe that the innovations of this study are: (i) we discovered a new channel to collect data—WeChat applet. Compared with traditional methods, collecting data through WeChat applet is better in cost, convenience and efficiency; (ii) we optimized the existing formula for calculating ovarian age to make it a suitable ovarian age assessment model for healthy Chinese women. This study was aimed to provide new insights for localization of the model and to prove its feasibility. We have revised the introduction, hoping that the new content will reflect our innovations better.

5. In order to recognize the validity and reliability of data collection or measurements, please describe this section much more than before.

Re: Thank you for your suggestion. To ensure the validity and reliability of data collection or measurements, we established strict criteria for collecting subjects’ data and screening valid samples. The details are as follows：

(i)Healthy women used to build the model training set, i.e. subjects recruited through CRO companies and WeChat applet, need to meet the following criteria: history of natural conception, intact ovaries, and normal menstruation, with an average interval of 21 to 35 days. The exclusion criteria were: estrogen or progesterone use two months before registration, breastfeeding, pregnancy, history of female infertility, endometriosis, ovarian cystic mass, history of ovarian surgery, PCOS, gynecological malignancy, history of radiotherapy or chemotherapy, autoimmune diseases, chronic systemic metabolic and endocrine diseases including hyperandrogenemia, hyperprolactinemia, diabetes, thyroid disease, hypogonadotropic hypogonadism, and drug usage that may cause irregular menstruation. These criteria were reflected in lines 119-126 of the previous manuscript.

(ii) The subjects in the test set included healthy women and women with PCOS and DOR. The inclusion criteria for healthy women are described in point one. The inclusion criteria for people with PCOS were at least 2 of the following 3 characteristics according to Rotterdam criteria: hypomenorrhea/amenorrhea, clinical and/or biochemical hyperandrogenemia, and polycystic ovarian morphology (PCOM). The inclusion criteria for women with DOR were: age ≥ 40 years old, FSH ≥ 12 mIU/mL, AFC ≤ 5~7, and AMH ≤ 1.1 ng/mL. Meeting two or more of the above four items was determined to be DOR. These were reflected in lines 127-134 of the previous manuscript.

(iii)Subjects recruited in all three ways were required to have test results for the following indicators: AMH, FSH, luteinizing hormone (LH), estradiol (E2), progesterone (P), testosterone (T), prolactin (PRL), and vitamin D levels of subjects from day 2 to day 4 of the menstrual cycle, pelvic ultrasound or vaginal ultrasound, height and weight on the day of the hormone test. It's worth noting that pelvic ultrasound or vaginal ultrasound was not required for subjects recruited from WeChat applet, but all other indicators were dispensable, otherwise the subject could not be included in the sample set. These were described in lines 90-93 of the previous manuscript, and we made some additions to them in this revision.

4. Subjects’ clinical test data obtained from hospital and CRO were diagnosed by clinicians, thus have reliability. Therefore, as long as subjects in these two sample sets meet the inclusion criteria, their data would be used. However, for the subjects’ data collected through the WeChat applet, we conducted a rigorous screening of these samples and eliminated some invalid ones, such as those with duplicate submissions, those with certain indicators very far from the criteria, and those with conflicting trends in certain indexes.

6. Which approaches were used for sampling?

7. The sample size is based on which formula?

6&7Re: Thank you for your question. Since the study is a retrospective study, the purpose of the study is not to investigate the relationship between PRL or AMH and age to establish a new relationship equation, but only to modify and optimize the existing equation. Therefore, we included all the collected data that satisfied the criteria in the training set. Due to the limited sample size of the data, we did not perform sampling based on the randomization principle as in general experimental designs, nor did we perform sample size calculations to ensure the power of statistical tests in order to utilize as much data as reasonably possible.

8. Please describe any efforts to address potential sources of bias.

Re: First of all, during the sample collection, we strictly control the quality of samples to exclude bias as much as possible. For hospital subjects and CRO subjects, their examination reports had to be diagnosed by clinicians to determine whether they met the inclusion criteria; for samples collected by WeChat applet, we screened and excluded invalid ones. Secondly, when improving the model, we referred to the formula for calculating ovarian age in previous literature, and we have taken into account the relevant factors that may be biased in the original formula and included them in the regression equation. We tried to eliminate as much bias as possible by the above two methods, but we have to admit that there may be some potential bias that we have not found or considered.

9. In this first paragraph of the discussion it is necessary to depict the main results of the study.

Re: Thank you for your suggestion. After group meeting, we did find that the original discussion section lacked the description of the main results and ideas, so we adjusted the first paragraph of the discussion section. The additions are as follows:

In this study, firstly, we developed an online ovage calculator based on the WeChat applet by combining the ovage calculation formula of Venturella R et al. The backend data showed that it could achieve the fastest collection of data for a population sample of 9000 cases in two weeks. Although these data were relatively raw and uncontrollable, it was still able to filter out 97 complete and compliant data, and the collection process is almost costless. There is no doubt that this method is more efficient, cheaper and safer than traditional methods of clinical data acquisition. Secondly, we collected a total of 262 cases of Chinese population data from hospital, CRO and WeChat applet, and divided them into training set and test set, and made an adaptation optimization of Chinese population data based on the ovage formula of Venturella R et al. We established a normal linear regression model with the independent variable-age, and used stepwise linear regression based on AIC values for variable selection. The optimized formula OvAge=exp (3.5254-0.0001*PRL-0.0231*AMH) is currently an appropriate model for ovarian age assessment in healthy Chinese women, and the accuracy of calculation is better than the original literature formula, which suggests that our optimization for the literature formula is feasible and effective. However, there is still a limitation that the accuracy of ovarian age prediction for DOR/PCOS population is low, which is mainly due to the limitation of sample size as only healthy population data are included in the training set, while the test set contains both healthy and disease population data. We will continue to optimize the ovarian age calculation formula using the same way to improve the accuracy of prediction for the Chinese disease population.

10. In discussion section lines (264-272, 306,) there is no citation or references.

Re: Thank you for pointing out the problem. It was indeed an oversight in the writing of our article. The corresponding references and explanations have now been added.

(i) Lines 275-276(Original manuscript, Lines 264-265) (revised manuscript with track changes), “In 2015, Ventureira and others proposed ‘Ovage’ to meet current requirements”, the cited literature is [1-5].

(ii) Lines 265-272(Original manuscript), “In previous studies, we found that in PCOS and DOR populations, the ovarian age calculated by the existing OvAge model was significantly smaller or greater than its actual age”. This argument has no reference, refers to the early stage of the study, it was concluded when we combined the pathological characteristics of the two disease populations [14,18] to make speculations and validated them with literature formulas (Table 2).

(iii) Line 306(Original manuscript), “This large-scale prospective study based on the Chinese population is of great significance for establishing an ovarian age prediction model based on the characteristics of Chinese women”. This argument refers to this study, but the original description is ambiguous. Since this study is a retrospective study and not a prospective study, we decided to revise the original text to “This retrospective study based on Chinese women has important implications for the development of an ovarian age prediction model based on Chinese female characteristics”.

11. Discuss limitations of the study, taking into account sources of potential bias or imprecision.

Re：Because the model in this study is based on the localized improvement and upgrade of the previous model-OvAge, we have referred to the relevant factors of the regression equation in the related document. We have to admit that our study does have shortcomings, such as the WeChat applet of collecting samples is indeed economical and convenient on one hand, but its data is not as real and reliable as the offline method on the other hand, which is also a point we need to consider and improve when collecting samples in our later study.

12. Discuss both the direction and magnitude of any potential bias.

Re：Because the model in this study is an improved version and optimization of the original formula for calculating ovarian age, so our model inherits some shortcomings of the original formula, which is something we can hardly change. In addition, our study does have some shortcomings, such as the low reliability of the sample data collected by the WeChat applet, and the need to screen the valid samples at a later stage, which may cause some bias in the study results if some invalid samples are mixed in. Therefore, it is especially important to conduct strict quality control on the samples collected by WeChat applets, which is something we need to pay attention to and improve in the later study.

13. The manuscript should be edited by a native English speaker.

Re：I'm deeply sorry that we have such mistake like language problems. The author's language skills are not very solid, which is why these problems exists, and I'm sorry for the bad reading experience. We are now looking for someone with a good command of English to adjust the language in the text systematically.

14. Please add the footnote of tables. The tables are chaotic.

Re：Thank you for pointing out the shortcomings of this manuscript. We have added footnotes to the picture and tables in the corresponding places of the revised manuscript. We have also re-edited the format of the tables.

Table 1. Participant data from Phase I.

Indicators Healthy Women (n=16) DOR Women (n=52) PCOS Women (n=45)

Chronological Age 28.44±2.99 29.27±3.76 28.33±2.83

AMH (ng/mL) 3.64±3.42 1.02±1.03 12.79±3.43

FSH (mIU/mL) 6.28±2.00 22.70±11.64 5.80±1.87

PRL (ng/mL) 14.28±5.13 12.43±4.51 10.72±4.78

AFC 5.63±2.09 3.42±1.39 9.14±4.37

BMI 23.39±4.22 23.36±2.81 26.58±6.41

Table 2. Comparison of chronological age and age calculated by OvAge in the literature.

Indicators Chronological Age Literature OvAge P value

Healthy women (n=16) 28.44±2.99 30.63±3.02 0.0818

DOR women (n=52) 29.27±3.76 31.29±5.42 0.0402

PCOS women (n=45) 28.33±2.83 26.55±2.46 0.0002

Table 3. Summary of model fit for Phase II.

Item Estimate Std. Error T value P value

Mod1a (Intercept) 32.3968 1.9030 17.0244 <0.0001

 PRL 0.1483 0.0872 1.7001 0.0955（-）

 AMH -0.8752 0.3296 -2.6557 0.0106（*）

Mod2a (Intercept) 3.4846 0.0568 61.4003 <0.0001

 PRL 0.0043 0.0024 1.7452 0.0810（-）

 AMH -0.0287 0.0103 -2.7813 0.0054（**）

Mod3a (Intercept) 32.2573 1.7938 17.9829 <0.0001

 PRL 0.1438 0.0867 1.6577 0.0974（-）

 AMH -0.8163 0.2971 -2.7474 0.0060（**）

Table 4. Comparison of fit results for phase Ⅱ.

Models MSE RMSE MAE

Mod1a 69.2735 8.3231 5.3830

Mod2a 117.1521 10.8237 5.9000

Mod3a 67.7729 8.2324 5.3772

Table 5. Comparison of chronological age, OvAge calculated age in the literature, and the established Mod3a calculated age.

Indicators Chronological Age Literature OvAge Mod3a P1 value (literature) P2 value（Mod3a）

Healthy women (n=68) 30.79±5.91 31.45±4.89 31.44±2.27 0.3984 0.3344

DOR women (n=52) 29.27±3.76 31.29±5.42 28.17±2.42 0.0402 0.1097

PCOS women (n=45) 28.33±2.83 26.55±2.46 28.66±1.60 0.0002 0.4984

Table 6. Summary of model fit for Phase Ⅱ and Phase Ⅲ.

Item Estimate Std. Error T value P value

Mod1b (Intercept) 33.8402 0.8489 39.8635 <0.0001

 PRL -0.0016 0.0072 -0.2253 0.8221（-）

 AMH -0.7045 0.1968 -3.5794 0.0005（***）

Mod2b (Intercept) 3.5254 0.0264 133.3673 <0.0001

 PRL -0.0001 0.0002 -0.2282 0.8195（-）

 AMH -0.0231 0.0064 -3.6403 0.0003（***）

Mod3b (Intercept) 33.6966 0.8367 40.2746 <0.0001

 PRL -0.0017 0.0071 -0.2407 0.8098（-）

 AMH -0.6608 0.1872 -3.5297 0.0004（***）

Table 7. Evaluation of model fit for Phase Ⅱ and Phase Ⅲ.

Models MSE RMSE MAE

Mod1b 32.0149 5.6582 4.1787

Mod2b 31.7237 5.6324 4.1703

Mod3b 32.6351 5.7127 4.2063

Table 8. Comparison of chronological age and three different models' calculated age.

Indicators chronological Age Literature OvAge Mod3a Mod2b P1 value (literature) P2 value (Mod3a) P3 value by (Mod2b)

Healthy women (n=68) 30.79±5.91 31.45±4.89 31.44±2.27 30.95±1.63 0.3984 0.3344 0.8169

DOR women (n=52) 29.27±3.76 31.29±5.42 28.17±2.42 28.28±1.74 0.0402 0.1097 0.1069

PCOS women (n=45) 28.33±2.83 26.55±2.46 28.66±1.60 27.63±1.10 0.0002 0.4984 0.4089

Figure Legends

Fig 1 Flowchart for optimizing the OvAge model.

Fig 2 Self-reporting system for subjects based on WeChat applet——the online OvAge calculator.

（A）QR code of online OvAge calculator；

（B）Main interface；

（C）Data Acquisition Module；

（D）Questionnaire module；

Table Legends

Table 1. Participant data from Phase I.

DOR：Diminished ovarian reserve. PCOS：Polycystic ovary syndrome. AMH：Anti-Müllerian hormone. FSH：Follicle-stimulating hormone. PRL：Prolactin. AFC：Antral follicle counting. BMI：Body mass index. n=16：16 sample data for healthy women. n=52：52 for DOR women. n=45：45 for PCOS women.

Table 2. Comparison of chronological age and age calculated by OvAge in the literature.

n=16：16 sample data for healthy women. n=52：52 for DOR women. n=45：45 for PCOS women.

Table 3. Summary of model fit for Phase II.

*** P < 0.001; ** P < 0.01; * P <0.05; - P>0.05.

Table 4. Comparison of fit results for phase Ⅱ.

MSE：mean squared error. RMSE：root mean squared error. MAE：mean absolute error.

Table 5. Comparison of chronological age, OvAge calculated age in the literature, and the established Mod3a calculated age.

n=68：68 sample data for healthy women. n=52：52 for DOR women. n=45：45 for PCOS women.

Table 6. Summary of model fit for Phase Ⅱ and Phase Ⅲ.

*** P < 0.001; ** P < 0.01; * P <0.05; - P>0.05.

Table 7. Evaluation of model fit for Phase Ⅱ and Phase Ⅲ.

MSE：mean squared error. RMSE：root mean squared error. MAE：mean absolute error.

Table 8. Comparison of chronological age and three different models' calculated age.

n=68：68 sample data for healthy women. n=52：52 for DOR women. n=45：45 for PCOS women.

15. Please discuss and compare the findings of this study with relevant literature much more than the current version.

Re：Thank you for your suggestion, and we have conducted more summary comparisons of the two studies in response to your suggestion. This whole study is intended to reflect an adaptive revision and upgrade of the original literature formula and is not intended to replicate an entirely new formula for calculating ovarian age exactly as the original literature study did. There are several differences between our study and those in the original literature (see the first paragraph of the Discussion for details), as follows.

(i) The type of sample chosen is different: the literature chose a population sample of Italian women, while we chose a population sample of Chinese women, and accordingly, the scope of application of both is different.

(ii) Different research aims: the original literature study applied the generalized linear model to the healthy fertility control dataset to derive the ovarian reserve function prediction formula, i.e., the ovage equation, while our study as a whole was based on the literature study and used the Chinese population data to screen the variables and reconfirm the regression coefficients, and finally the original formula was adapted and optimized for the Chinese population.

(iii) The sample size and modeling approach are different: the original literature study used a generalized linear model with a sample size of 710 cases of population data, while this study used a normal linear regression model with a sample size of 262 cases of population dataset.

(iv) The original study was designed to explore clinical, biochemical and 3D ultrasound data to establish a simple, intuitive and easy to understand method of defining ovarian reserve. The accuracy of results analysis is different.

(v) The accuracy of the results analysis is different: the original study results can predict the ovarian reserve level of healthy, POI, PCOS sub-healthy population more accurately, the study is in the early stage of research so it can predict the ovarian reserve level of healthy population more accurately, for PCOS and DOR population need to be further explored.

Reviewer 2:

1. The abstract was not representative of the study. Large number of contents regarding disease population (DOR/PCOS) were include in methods and results but was not mentioned at abstract. Please ensure that abstract and study contents are consistent.

Re：We are sorry that the abstract in our first draft did not mention this. Taking into account your comments and those of another reviewer, we have rewritten the abstract, and this comment is reflected in the revised abstract, which we hope will present our results well.

2. The optimized ovarian age calculation formula, mod2b, is more accurate than the literature OvAge formula in evaluating ovarian age of healthy patient population. However, the real value of a scientific model is to accurately predict disease population. This is the weak point of this study that weakens the value of this study.

Re：The issues you raised are also the main limitations we have noticed at this stage of the study (Word-revised manuscript with track changes, lines 227-235). As you said, the real value of a scientific model is to accurately predict those potential patients, but we were not able to collect enough data on disease populations to build a training dataset for those people due to the limited manpower, resources, time, and promotion strategy of the app. At the same time, because the dataset was built shortly after the ovage app was officially launched for public testing (only one month later), and most of the people invited for testing during the initial promotion of the app were healthy people.As a result,although we collected more than 9,000 people's data, there was still no data on the disease population that met the calculation requirements, and there were only 97 cases of healthy people (about 1% of the total population). So we did our best to optimize the Chinese population adaptation of the literature formula based on the existing data, although the new formula we obtained after the adjustment is not accurate for the prediction of the disease population, but the accuracy of the prediction of the healthy population has been improved compared with the original formula, which is an inevitable trend at this stage of the study and also proves to some extent that our optimization strategy is feasible. In the next phase of the study, we will continue to collect data from the disease population through the ovage app and use the same optimization method to optimize the formula to improve the accuracy of predicting ovarian reserve function of those patients.

2. The optimized ovarian age calculation formula, mod2b, is more accurate than the literature OvAge formula in evaluating ovarian age of healthy patient population. However, the real value of a scientific model is to accurately predict disease population. This is the weak point of this study that weakens the value of this study.

Re：The issues you raised are also the main limitations we have noticed at this stage of the study (lines 227-235). As you said, the real value of a scientific model is to accurately predict those potential patients, but we were not able to collect enough data on disease populations to build a training dataset for those people due to the limited manpower, resources, time, and promotion strategy of the app. At the same time, because the dataset was built shortly after the ovage app was officially launched for public testing (only one month later), and most of the people invited for testing during the initial promotion of the app were healthy people.As a result,although we collected more than 9,000 people's data, there was still no data on the disease population that met the calculation requirements, and there were only 97 cases of healthy people (about 1% of the total population). So we did our best to optimize the Chinese population adaptation of the literature formula based on the existing data, although the new formula we obtained after the adjustment is not accurate for the prediction of the disease population, but the accuracy of the prediction of the healthy population has been improved compared with the original formula, which is an inevitable trend at this stage of the study and also proves to some extent that our optimization strategy is feasible. In the next phase of the study, we will continue to collect data from the disease population through the ovage app and use the same optimization method to optimize the formula to improve the accuracy of predicting ovarian reserve function of those patients .

3. A reasonable experimental design is needed. The accuracy of ovarian age prediction for disease patient population (PCOS and DOR) was significantly worse than that of the literature OvAge formula. The underlying reason is that training data (phase II & III) used in this study from the first and second optimization were only from healthy people. If this is the case, prediction outcomes from such a model are unreliable. The whole manuscript needs to be re-written.

Re：Thank you very much for your question about our experimental design, which is pertinent and critical. This is an unnecessary misunderstanding due to our unclear description of the focus of our study, and we have integrated all the questions about the focus and made corresponding adjustments to the abstract, introduction and discussion sections of this paper to reduce unnecessary misunderstandings.

In general, there are only two main objectives of this study: (i) to develop a more efficient, inexpensive, and safe method of population data acquisition, which is not only a feasible solution to the difficulty of population clinical data collection, but also an attempt to combine IOT technology with the establishment of a new self-reporting system, and a safe strategy for initial patient self-diagnosis in the context of today's epidemic. (ii) In the present study, we have been able to find a feasible solution for ovage and other existing equations for ovarian reserve assessment that can be adapted and optimized for different regional populations. So, in this study about the exploration process we made in the model optimization scheme there are some misleading experimental design, we used a test set beyond the training set covering the sample type to test the prediction accuracy of the formula, i.e., the healthy population data was chosen as the training set, while the test set lacked the inclusion of disease population data, we made this design for three reasons: (i) Since data of the patient crowd is relatively low,it is not enough for training or testing. (ii) The main purpose of the experimental design is to explore an effective optimization strategy for the equation. (iii) The optimization only aims to improve the accuracy of the healthy population, and the inclusion of a small amount of disease population into the test dataset is only for the preliminary understanding of the deviation of the optimized model from the prediction results of the disease population, which can be used to guide the subsequent research.

4. The collected data were divided into three datasets: training set, test set, and prediction set.

Different populations (healthy or DOR, PCOS) differed in each dataset. I would suggest the authors to focus on model establishment of healthy Chinese women instead of trying to cover both populations.

Re：Your question and suggestions are very accurate. There are two reasons why we did not include disease data in the training set but included disease population data in the test set, which ultimately led to the low accuracy of disease population prediction: the first reason is the limitation of research resources and conditions; second is the purpose of this study. The purpose of this study is to illustrate two key points: (i) the ovage applet we created is an inexpensive and efficient data collection tool; (ii) as mentioned in your question, the application of the formula in this study focuses on the significant improvement in the prediction accuracy of the formula for ovarian age in healthy Chinese women through our optimization, which validates our formula optimization method to some extent. We will continue to improve the prediction accuracy for the disease population in the subsequent study by supplementing the data of the disease population.

5. Page 24 Line 265-267: Please provide references of the previous studies.

Re：Your concern is very detailed and the lack of clarity here is an oversight on our part. Lines 265-272 "In previous studies, we found that in PCOS and DOR populations, the ovarian age calculated by the existing OvAge model was significantly smaller or greater than its actual age.”This script means that in the previous period of the study, we combined the pathological characteristics of the two disease populations [14,18] to make speculations while validating them with literature formulas (Table2) to reach our conclusions at the same time.

---

## [Decision Letter · Decision Letter 1]

12 Dec 2022

Development, promotion, and application of online OvAge calculator based on the WeChat applet: Clinical prediction model research

PONE-D-22-09299R1

Dear Dr. Ren,

We’re pleased to inform you that your manuscript has been judged scientifically suitable for publication and will be formally accepted for publication once it meets all outstanding technical requirements.

Kind regards,

Jing Zhang

Academic Editor

PLOS ONE

**Comments to the Author**

1. If the authors have adequately addressed your comments raised in a previous round of review and you feel that this manuscript is now acceptable for publication, you may indicate that here to bypass the “Comments to the Author” section, enter your conflict of interest statement in the “Confidential to Editor” section, and submit your "Accept" recommendation.

Reviewer #1: All comments have been addressed

Reviewer #2: All comments have been addressed

2. Is the manuscript technically sound, and do the data support the conclusions?

Reviewer #1: Yes

Reviewer #2: Yes

3. Has the statistical analysis been performed appropriately and rigorously? 

Reviewer #1: Yes

Reviewer #2: Yes

4. Have the authors made all data underlying the findings in their manuscript fully available?

Reviewer #1: Yes

Reviewer #2: Yes

5. Is the manuscript presented in an intelligible fashion and written in standard English?

Reviewer #1: Yes

Reviewer #2: Yes

6. Review Comments to the Author

Reviewer #1: The authors have adequately addressed the comments raised in a previous round of review and I feel that this manuscript is now acceptable for publication.

Reviewer #2: The authors made meticulous revision and significantly improved the manuscript. My concerns have been addressed and there is no further question from me.

7. PLOS authors have the option to publish the peer review history of their article (what does this mean?). If published, this will include your full peer review and any attached files.

Reviewer #1: **Yes: **Fahimeh Ramezani

Reviewer #2: **Yes: **Yimin Shu

---

## [Editor Report · Acceptance letter]

15 Feb 2023

PONE-D-22-09299R1 

Development, promotion, and application of online OvAge calculator based on the WeChat applet: Clinical prediction model research 

Dear Dr. Ren:

I'm pleased to inform you that your manuscript has been deemed suitable for publication in PLOS ONE. Congratulations! Your manuscript is now with our production department. 

Kind regards, 

on behalf of

Dr. Jing Zhang 

Academic Editor

PLOS ONE